# Dpp/TGFβ-superfamily play a dual conserved role in mediating the damage response in the retina

Joshua Kramer[1,2], Joana Neves[2,3], Mia Koniikusic[2], Heinrich Jasper[2,4]*, Deepak A. Lamba[1,2]*

1 Department of Ophthalmology, University of California, The Eli and Edythe Broad Center of Regeneration Medicine and Stem Cell Research, San Francisco, CA, United States of America, 2 Buck Institute for Research on Aging, Novato, CA, United States of America, 3 Faculdade de Medicina, Instituto de Medicina Molecular (iMM), Universidade de Lisboa, Lisbon, Portugal, 4 Immunology Discovery, Genentech, Inc., South San Francisco, CA, United States of America

* deepak.lamba@ucsf.edu (DAL); jasperh@gene.com (HJ)

**Data Availability Statement:** All relevant data are within the manuscript and its Supporting Information files.

## Abstract

Retinal homeostasis relies on intricate coordination of cell death and survival in response to stress and damage. Signaling mechanisms that coordinate this process in the adult retina remain poorly understood. Here we identify Decapentaplegic (Dpp) signaling in *Drosophila* and its mammalian homologue Transforming Growth Factor-beta (TGFβ) superfamily, that includes TGFβ and Bone Morphogenetic Protein (BMP) signaling arms, as central mediators of retinal neuronal death and tissue survival following acute damage. Using a *Drosophila* model for UV-induced retinal damage, we show that Dpp released from immune cells promotes tissue loss after UV-induced retinal damage. Interestingly, we find a dynamic response of retinal cells to this signal: in an early phase, Dpp-mediated stimulation of Saxophone/Smox signaling promotes apoptosis, while at a later stage, stimulation of the Thickveins/Mad axis promotes tissue repair and survival. This dual role is conserved in the mammalian retina through the TGFβ/BMP signaling, as supplementation of BMP4 or inhibition of TGFβ using small molecules promotes retinal cell survival, while inhibition of BMP negatively affects cell survival after light-induced photoreceptor damage and NMDA induced inner retinal neuronal damage. Our data identify key evolutionarily conserved mechanisms by which retinal homeostasis is maintained.

## Introduction

A functional and resilient visual system, durable to potential insults, is crucial for rapid interpretation of an animal's surroundings. In both mammals and insects, the visual system contains a specific organization of structures; a lens to focus incoming light, a fluid filled vitreous, photoreceptors to sense and transmit light sensing signals to the optic lobes, pigmented cells to prevent diffraction, and immune cells adjacent to the tissue to mediate the damage response [1]. Photoreceptors that make up these systems become postmitotic in early maturation (24

**Funding:** The research presented here is supported by NIH grants (R01 EY025779 and EY032197 to DL; AG057353 and EY018177 to HJ; P30 Vision Core grant to UCSF Dept of Ophthalmology), and the Research to Prevent Blindness (unrestricted grant to UCSF Dept of Ophthalmology). The funders had no role in study design, data collection and analysis, decision to publish, or preparation of the manuscript. Dr. Jasper's affiliation with Genetech, Inc provided support in the form of salaries for author HJ, but did not have any additional role in the study design, data collection and analysis, decision to publish, or preparation of the manuscript. The specific roles of these authors are articulated in the 'author contributions' section.

**Competing interests:** I have read the journal's policy and the following authors of this manuscript have the following competing interests: Heinrich Jasper is an employee of Genentech Inc. This commercial affiliation does not alter author's adherence to all PLOS ONE policies on sharing data and materials.

hours after puparium formation (APF) in *Drosophila* [2]; the first postnatal week in mice [3,4], and no stem cell pool exists to replace them. As such, progressive loss of these cells due to injury can significantly impact the organ's function. When the eye suffers damage, retinal tissue must strike a balance between repair, survival, and apoptosis. This homeostatic response is mediated in part by immune cell-derived cytokines and growth factors, which influence the balance between apoptosis and survival of damaged cells [5–7]. The exact interplay of these factors in the retinal damage response remains to be fully understood.

*Drosophila* is an excellent model for studying the interaction between immune cells and photoreceptors during retinal tissue repair. 90% of *Drosophila* hemocytes are macrophages (plasmatocytes), existing in clusters around every major organ including the eyecups [8]. Genetic or UV-C induced damage directed to the post-mitotic *Drosophila* pupal retina can induce reproducible, quantifiable damage that persists to adulthood and is sensitive to genetic perturbations, allowing dissection of pathways that mediate photoreceptor apoptosis and control survival [2,7,9]. Persistent DNA damage induced by UV has been shown to promote initiator caspase activity in photoreceptors and thus apoptosis [10–17].

After UV damage, hemocytes are recruited to the retina, where they are activated by the Pdgf1 orthologue Pvf1 in response to activation of the Dpp signal transducer Schnurri (Shn) in damaged retinal cells [9,18]. Pvf1, in turn, induces the neurotrophic factor Mesencephalic Astrocyte Derived Neurotrophic Factor (MANF), in hemocytes, regulating their transition to an anti-inflammatory, pro-repair phenotype [7] and this response is conserved in mice [7]. However, the precise role of Dpp/Shn signaling in coordinating the tissue repair response remains unclear.

Canonically, Dpp signals through two downstream type 1 receptors, Thickveins (Tkv) and Saxophone (Sax), which both form heterotetrameric complexes with the type II receptor Punt [19]. The downstream target of Tkv is Mad, while Sax phosphorylates Smox, resulting in its nuclear translocation [20]. A coordination of Tkv and Sax signaling has recently been described in the tissue damage response of the fly intestinal epithelium, where Sax/Smox signaling mediates activation of intestinal stem cell (ISC) proliferation in response to enteropathogen infection, while Tkv/Mad signaling promotes the return to quiescence of these cells [21]. This response is coordinated by the control of Tkv protein turnover in ISCs. Tkv degradation is reduced during ISC activation, allowing for Tkv protein accumulation and replacement of Sax in Tkv/Punt complexes [22].

In mammals, Dpp homologues include various members of the bone morphogenic protein (BMP) and transforming growth factor beta (TGFβ) family. Their downstream targets include Smad 1/5/9 (activated by BMP) and Smad 2/3 (activated by TGFβ). BMP 2/4 have been shown to act in an anti-inflammatory manner in multiple systems, biasing macrophages to their anti-inflammatory M2 subtype [23–26]. These proteins also stimulate Müller glia proliferation, promote survival of retinal ganglion cells after damage, and decrease microglial activation *in vivo* [27–30]. However, the timing and mechanism(s) by which Dpp/BMP signaling modulates repair in the retina is not fully understood. It is also unclear how these two homologous pathways interact in this tissue.

In this report, we characterize the role of Dpp/BMP/TGFβ signaling in the retinal damage response of *Drosophila* and mice. We find a conserved role for early Sax/Smox signaling and Smad 2/3 in promoting photoreceptor apoptosis after light damage, and a later role for Tkv/Mad and Smad 1/5/9 signaling in promoting survival. We further provide evidence for a role of immune cells as sources for the ligands of these pathways after injury. Our findings provide critical new insight into mechanisms that maintain retinal homeostasis after injury.

## Results

### Hemocyte derived Dpp controls the damaged retinal tissue response in *Drosophila*

Our previous findings had identified Shn as a critical mediator of the retinal damage response in flies (9), yet the role of Dpp signaling in that response remained unclear. To start assessing a possible role for Dpp signaling components in retinal apoptosis, we performed genetic interaction experiments using a previously characterized fly line overexpressing a constitutively active form of the Jun N-terminal Kinase Kinase (JNKK) Hemipterous (Hep) under the control of the photoreceptor and cone cell driver Sep-Gal4 [9,31–34]. In this line (Sep-Hep^ACT^), Hep^ACT^ expression is initiated during development in postmitotic photoreceptor and cone cells in the third instar larval eye disc, inducing photoreceptor apoptosis through activation of the pro-apoptotic gene *hid* [32,35].

When Sep-Hep^ACT^ was crossed to flies expressing Sax-RNAi or Smox-RNAi under the control of UAS, a significant increase in surviving photoreceptors compared to control flies (crossed to UAS::mCherry-RNAi) was observed (Fig 1A and 1A'). Overexpression of the negative feedback inhibitor of Dpp signaling, Daughters against Dpp (Dad; [36,37]), also promoted survival when compared to control (Figs 1A' and S1F). Conversely, we found that knocking down Tkv and Mad resulted in a significant decrease in surviving photoreceptors (Fig 1A and 1A').

To assess whether this antagonistic effect of Sax/Smox and Tkv/Mad signaling are also observed during UV-induced cell death, we used an assay in which the headcase of pupae is removed at 24 hours after puparium formation, and animals are irradiated on one side with 17.5 microjoules of UV-C radiation [2,32]. In this model, hemocytes are attracted to the retina after UV-C induced damage and are critical to limit excessive cell death. When animals emerge after development, the extent of apoptosis can be measured by quantifying the size of the irradiated eye in relation to the non-irradiated control eye. We used the glass multimer reporter driver (GMR-Gal4; [38]) to express RNAi targeting Dpp pathway components in all postmitotic cells of the retina, and found that Sax or Smox knockdown inhibits UV-induced tissue loss, while Tkv or Mad knockdown promote it (Fig 1B and 1B'). A similar protective effect was observed when we knocked down Dpp in hemocytes used a hemocyte-specific driver (hemolectin::Gal4; [39]) (Fig 1B) in support of previous studies from our lab and others demonstrating that hemocytes secrete Dpp [21,40–44]. These data further support a model in which Sax/Smox signaling promotes apoptosis and Tkv/Mad signaling inhibit apoptosis in the retina.

To directly investigate the effects of Dpp signaling in photoreceptors, we irradiated *Drosophila* pupa, waited an additional 24 hours as previously described [2,9], and analyzed retinae during the post maturation stage (27, 30, 36 and 48 hours post UV exposure; equivalent to 51, 54, 60 and 72 hours post puparium formation). We used expression phospho-Mad (pMad) to assess activation of the Tkv/Mad pathway in Elav+ photoreceptors [45] during each post maturation timepoint, under both non UV control (S1A and S1A' Fig) and UV condition (Fig 1C). After UV treatment, pMad activity in Elav+ cells peaked at 36 hours (Fig 1C'). We further assessed Smox nuclear localization using a Smox::GFP knock-in line in which GFP-tagged Smox is expressed from the endogenous Smox locus [46] in non UV control condition (S1B and S1B' Fig). Nuclear localization of Smox represents activation of the Sax/Smox response [21]. Compared to pMad, Smox undergoes nuclear translocation at the earliest time point we assessed, 27 hours post UV exposure (Fig 1D and 1D'). To assess whether the kinetics of Sox/Smox and Tkv/Mad activation correlates with the induction of apoptosis, we stained for cleaved *Drosophila* caspase-1 (DCP1), a short-prodomain caspase and crucial driver of cell

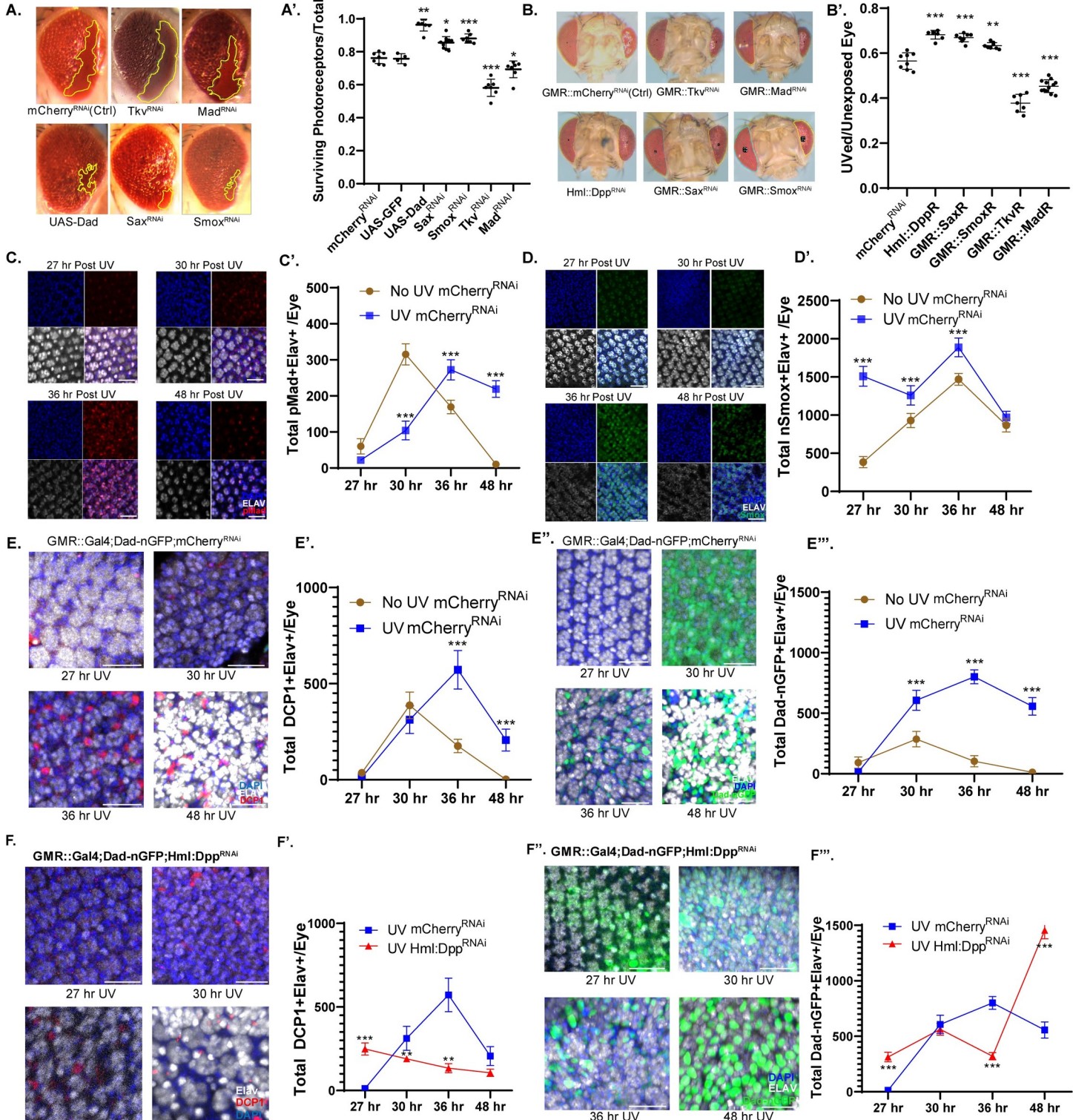

**Fig 1. Dpp inhibition controls JNK and UV driven apoptosis in Drosophila.** (1A-A') Representative images and quantitation of ratio of surviving photoreceptors to total in w;Sep::Gal4,UAS::HepACT flies crossed with RNAi lines of each genotype. Representative images and quantification of adult GMR::Gal4; /UAS::RNAi progeny following knockdown of is compared through ratio of UV-exposed to unexposed eye (1B-B'). Representative images and quantitation of the pMad (in maturing GMR::Gal4;Dad-nGFP/UAS::RNAi) and nuclear Smox (in GMR::Gal4;Smox::FLAG::GFP/UAS::RNAi) (1D-D') expression in Elav+ (1C-C') progeny eye with and without irradiation at 27, 30, 36 and 48hours post UV. Representative images and quantitation of total DCP1+ Elav+ cells (1E-E') or DadnGFP+/Elav+ cells (1 E"-F"') in GMR::

Gal4;Dad-nGFP/UAS::mCherry$^{RNAi}$ progeny with and without irradiation at various time points post exposure. Representative images and quantitation of total DCP1 + Elav+ cells (1FF') or DadnGFP+Elav+ cells (1 F''-F''') in maturing GMR::Gal4;Dad-nGFP/ Hml::Dpp$^{RNAi}$ progeny compared to controls. Scale Bar: 20 μm. Experiments were conducted with n = 2 replicates and n = 3–9 eyes/timepoint. Error bars indicate s.e.m. P-values from Student's t-test. *p<0.05, **p<0.01, **p<0.001.

death [47]. When compared to control animals, DCP1 activity peaks at 36 hours post UV exposure (Fig 1E and 1E'). Expression of Dad as reported by the DadnGFP reporter [48,49] also peaked in Elav+ cells at 36 hours following irradiation (Fig 1E" and 1E''').

These kinetics led us to hypothesize that Mad activation induced Dad transcription inhibits Sax/Smox signaling post-UV thereby limiting apoptosis. To test this, we examined the effects of Dpp, Tkv/Mad and Sax/Smox perturbations on UV-induced apoptosis. Knocking down Dpp in immune cells significantly reduced apoptosis (as determined by anti-DCP1 staining) when compared to controls (Fig 1F and 1F'), while also resulting in a significant increase in Dad activity at 48hours post UV (Fig 1F" and 1F''').

Knockdown of Dpp in immune cells under non UV state did not significantly differ from control level of apoptosis (S1C and S1C' Fig). Knockdown of Sax or Smox, on the other hand, resulted in a significant decrease in apoptosis at the 30 and 36 hour time points (Fig 2A, 2A', 2B and 2B') compared to control, associated with a significant increase in Dad-nGFP expression (Fig 2A"–2B")(S3A, S3A'–S3B, S3B' Fig). Knocking down Tkv or Mad, in turn, significantly increased the number of DCP1+ cells at every time point (Fig 2C, 2C'–2D, 2D'), and decreased Dad-nGFP reporter activity (Figs 2C"–2D"; S3C, S3C'–S3D, S3D'). These observations were recapitulated when Tkv/Mad and Sax/Smox perturbations were targeted directly to photoreceptors and cone cells by Sep::Gal4 rather than to the whole retina (Figs 2E, 2E'–2F, 2F' and S4). DCP1+ cell count under control Sep::Gal4 had similar results to GMR::Gal4 under UV and non UV states (S3D, S3D' and S3D" Fig).

Our data suggest a bimodal response of photoreceptors to Dpp after UV irradiation that is controlled by Sax and Tkv signaling. Additionally, suppression of Dad expression by Smox and subsequent activation of Dad expression by Mad contribute to the regulation of photoreceptor cell death and survival.

## BMP and TGFβ signaling are activated in a time dependent manner following retinal stress and damage

As the *Drosophila* visual system has many similarities with the mammalian retina [50], we next asked whether the bimodal response of the two arms of the Dpp pathway to damage in the *Drosophila eye* is conserved in mammals. We specifically focused on the TGFβ superfamily, which has a close homology to the *Drosophila* Dpp canonical pathway and has shown to play a role in the immune response in multiple systems [51,52]. Two important members of the family, BMP and TGFβ, share close homology with Dpp [52,53]. Following binding-induced TGFβ type 1 receptor phosphorylation, they trigger phosphorylation and nuclear translocation of Smads (Smad 1/5/9 for BMP pathway or Smad 2/3 for TGFβs), and transcriptional activation of Smad target genes. Based on our studies in *Drosophila*, we hypothesized that the TGFα/Smad2/3 arm of the pathway is responsible for the inflammatory or apoptotic response, and the BMP/Smad1/5/9 arm for the anti-inflammatory pro-repair response.

To test this hypothesis, and to elucidate the dynamics of TGFβ superfamily activation, we used light-induced photoreceptor stress and damage models previously used by various labs including ours [7,54,55]. Briefly, C57BL/6 mice were exposed to high-intensity (10,000 lux) light for 1.5 hours to induce retinal stress without overt retinal apoptosis. We then analyzed the phosphorylation of the mammalian homologues to Mad and Smox, Smad1/5/9 and Smad2/3 [56] comparing non-light exposed conditions to control (S5A Fig), 0, 6, 12, 24 and 36

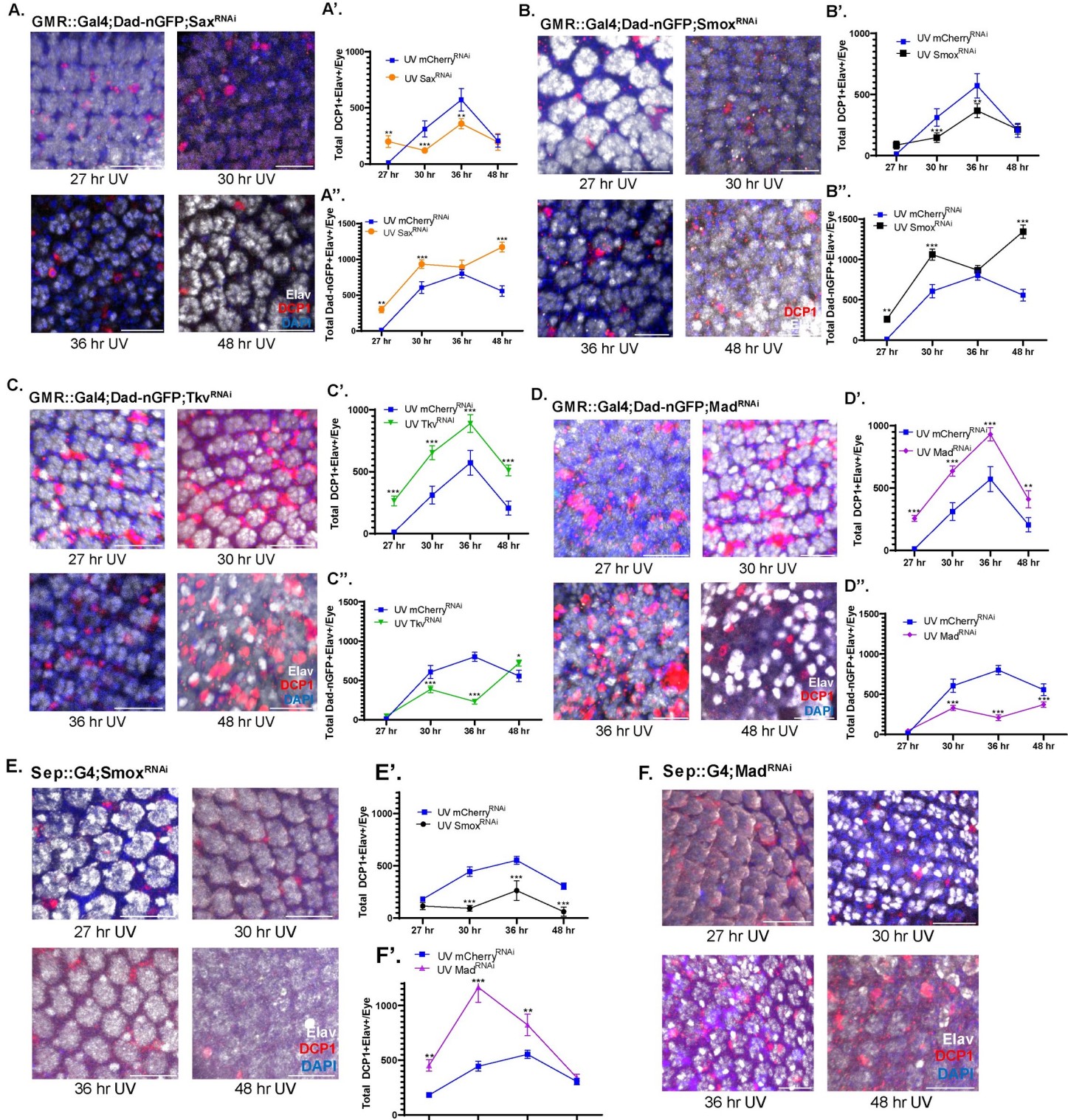

**Fig 2. Following UV driven whole eye radiation damage, Sax and Smox RNAi inhibit apoptosis, while Mad and Tkv RNAi promote it.** Representative images and quantitation of total DCP1+ Elav+ cells (2-A'; 2B-B') or DadnGFP+ Elav+ cells (2A"-S2A"'; 2B"-S2B"') in maturing GMR::Gal4;Dad-nGFP/ Sax-RNAi and Smox-RNAi progeny compared to controls. Representative images and quantitation of total DCP1+ Elav+ cells (2C-C'; 2D-D') or DadnGFP+ Elav+ cells (2C"-S2C"'; 2D"-S2D"') in maturing GMR::Gal4;Dad-nGFP/ Tkv-RNAi and Mad-RNAi progeny compared to controls. Representative images and quantitation of total DCP1+ Elav+ cells (2-E-E'; 2F-F') in maturing Sep::Gal4;/ Smox-RNAi and Mad-RNAi progeny compared to controls. Scale Bar: 20 μm. Experiments were conducted with n = 2 replicates and n = 3–5 eyes/timepoint. Error bars indicate s.e.m. P-values from Student's t-test. *p<0.05, **p<0.01, **p<0.001.

hours post exposure. Upon exposing 3.5 month old C57BL/6 mice to light, we observed elevated levels of phosphorylated Smad 1/5/9 (BMP response) in the inner nuclear layer (INL) of the retina, starting at 6 hours post-light exposure, and peaking at 12 hours post exposure (Fig 3A and 3A'). Phosphorylation of Smad 2/3 (TGFβ activation) peaked earlier, at 6 hours post exposure (Fig 3B and 3B').

Next, we asked if the response differs in BALB/c mice, which are more sensitive to light damage and exhibits widespread apoptosis in the retina following light exposure [54,57]. BALB/c mice were exposed to 5,000 lux light for 1 hour to induce apoptosis as previously shown [7]. TUNEL assay was performed on retina 24 hours post exposure to confirm significant apoptosis (S5B Fig). Interestingly, even prior to light exposure, BALB/c mice exhibit persistent TGF-β activity (pSmad 2/3) in the inner nuclear layer, suggesting substantial retinal stress under normal light conditions (Fig 3C). Immediately after light exposure (0 hours), this activity increases transiently, but by 6 hours, the pSmad2/3 signal returns to pre-light exposure levels and is absent once apoptosis sets in at 24 hours (Fig 3C'). As opposed to C57BL/6 mice, BALB/c mice exhibited no phosphoSmad1/5/9 (BMP) activity in the inner retina either under control conditions or after light exposure at any of the timepoints assessed in our study (S5D Fig). These studies suggest that TGFβ signaling may be involved in the early response which is typically of inflammatory nature, while BMPs may play a role in a later response which tends to be protective and anti-apoptotic. The absence of BMP/SMAD1/5/9 response in BALB/c mice was further investigated below to test a cause or effect conundrum.

We and others have previously shown that the retinal damage response relies on factors secreted by activated immune cells [58–61]. We next tested whether the light stress-induced BMP response is impacted when immune cells are absent. We used CD11b::DTR mice wherein myeloid cells can be inducibly ablated using intraperitoneally injected diphtheria toxin (DT) [62,63]. We have previously shown that light stress results in photoreceptor apoptosis in CD11b::DTR mice despite them being on C57BL/6 background [7]. DT induced immune cell loss in CD11b::DTR mice also resulted in loss of pSmad 1/5/9 in the inner retina under apoptosis inducing light-damage condition (Fig 3D and 3D' in comparison to PBS injected light damaged control retinas where pSmad 1/5/9 activation was seen in CRALBP + Müller glial cells in (S5E Fig). These studies suggest that immune cells participate in the damage-induced BMP response in mammalian retinas.

## Modulation of the BMP signaling pathway protects against damage induced retinal apoptosis

We hypothesized that BMP mediates an anti-inflammatory protective retinal damage response and sought to test this hypothesis by asking whether modulation of BMP signaling would alter retinal repair response. We used the NMDA excitotoxic damage model previously used by multiple labs to cause inner retinal damage [64–67] in addition to the light damage model. The excitotoxin NMDA damages the ganglion and amacrine cells in the retina and has been used to model retinal damage due to glaucoma, retinal ischemia and diabetic retinopathy [68–71]. To directly monitor BMP activity, we used BRE-eGFP transgenic mice. These mice report the transcriptional response of BMP-Smad activation through the BMP response element (BRE) [72]. In these mice, GFP expression was observed in CRALBP+ Müller glia following NMDA exposure (Fig 4A).

To modulate the BMP signaling pathway, mice were injected intravitreally with either recombinant BMP4 or the small molecule BMP inhibitor Dorsomorphin [28,73]. Upon comparing various treatments, we confirmed that GFP activity increases following BMP4 treatment and reduces with Dorsomorphin (Fig 4A and 4A'). TUNEL staining at 48-hours post-

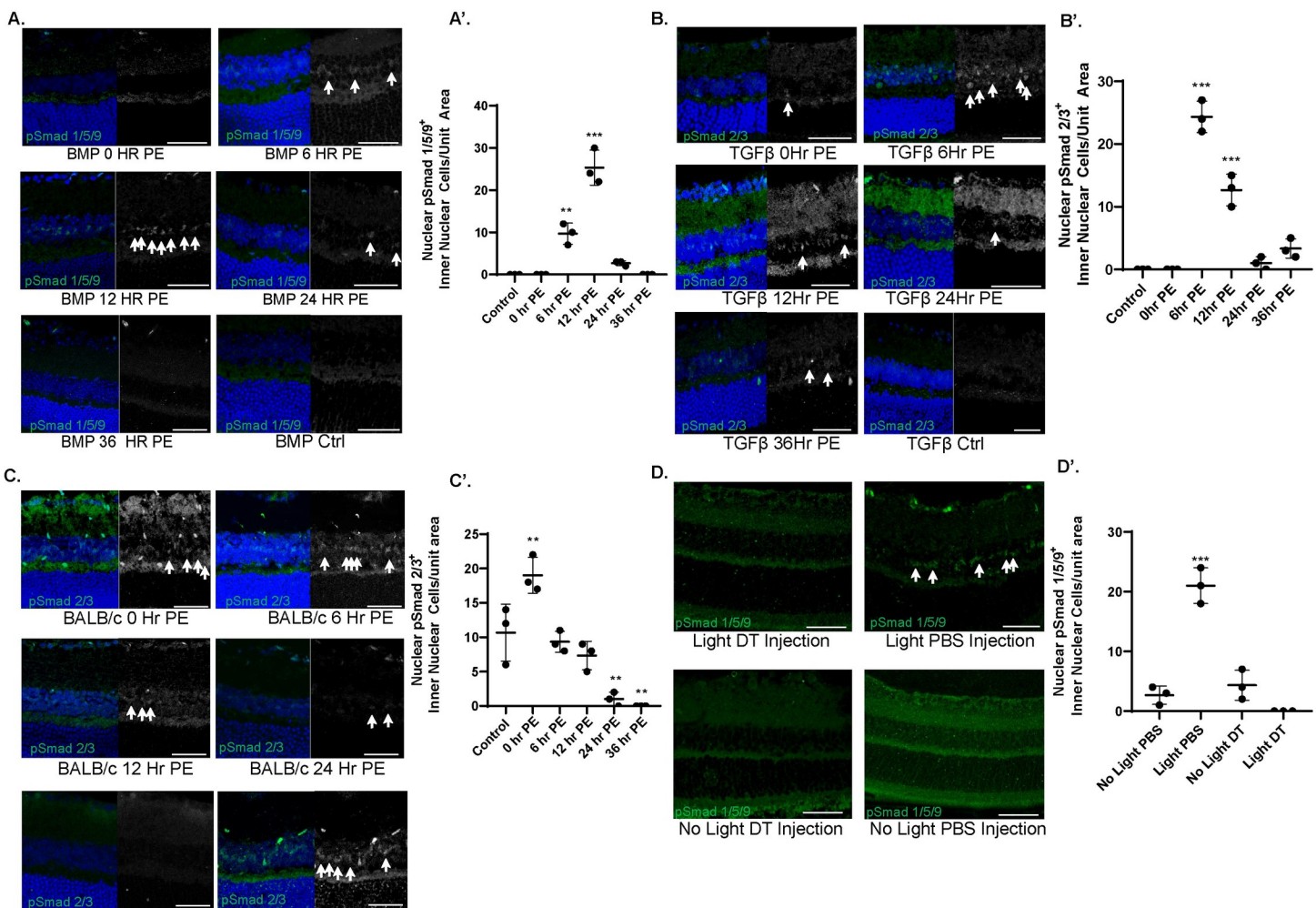

**Fig 3. BMP and TGFβ activation occurs in a time dependent manner following light stress and damage.** Representative retinal images and quantitation of control dark adapted 3.5 m/o C57BL/6 mice, or 0, 6, 12, and 24 hours post exposure to 10,000 lux for 1.5 hours stained for pSmad 1/5/9 (in green, 3A-A') and pSmad 2/3 (in green,3B-B'). Similar analysis of pSmad 2/3 in light-damaged BALB/c mice retinal sections (3-C). Representative retinal images and quantitation in 3.5 m/o CD11b::DTR mice treated with either diphtheria toxin or PBS control and exposed to 10,000 Lux. Sections were stained pSmad 1/5/9 (green) (3D-D'). Arrows (red) highlight pSmad expression in the inner retinal in all images. Scale Bar: 30 μm. Experiments were conducted with n = 2 replicates and n = 3–5 mice/condition. Error bars indicate s.e.m. Pvalues from Student's t-test. *p<0.05, **p<0.01, **p<0.001. PE = post-exposure. DAPI (blue) marks nuclei in all panels.

NMDA injection revealed that inhibition of BMP signaling increased apoptosis, while supplementation of BMP4 inhibited NMDA induced apoptosis (Fig 4B and 4B'). To assess whether this was a conserved retinal damage response or specific to NMDA damage, we assessed pathway modulation under light-induced photoreceptor damage conditions (20,000 Lux for 2 hours [74]). TUNEL staining at 48 hours post exposure revealed that inhibition of BMP signaling increases apoptosis and supplementation of recombinant BMP4 reduces it (4C,C'), similar to our observations in the NMDA damage model.

## Inhibition of TGFβ signaling pathway protects against damage induced retinal apoptosis

Since our data suggested that TGFβ may be involved in the earlier inflammatory response to damage, we tested if inhibiting the pathway using the small molecule TGFβ inhibitor

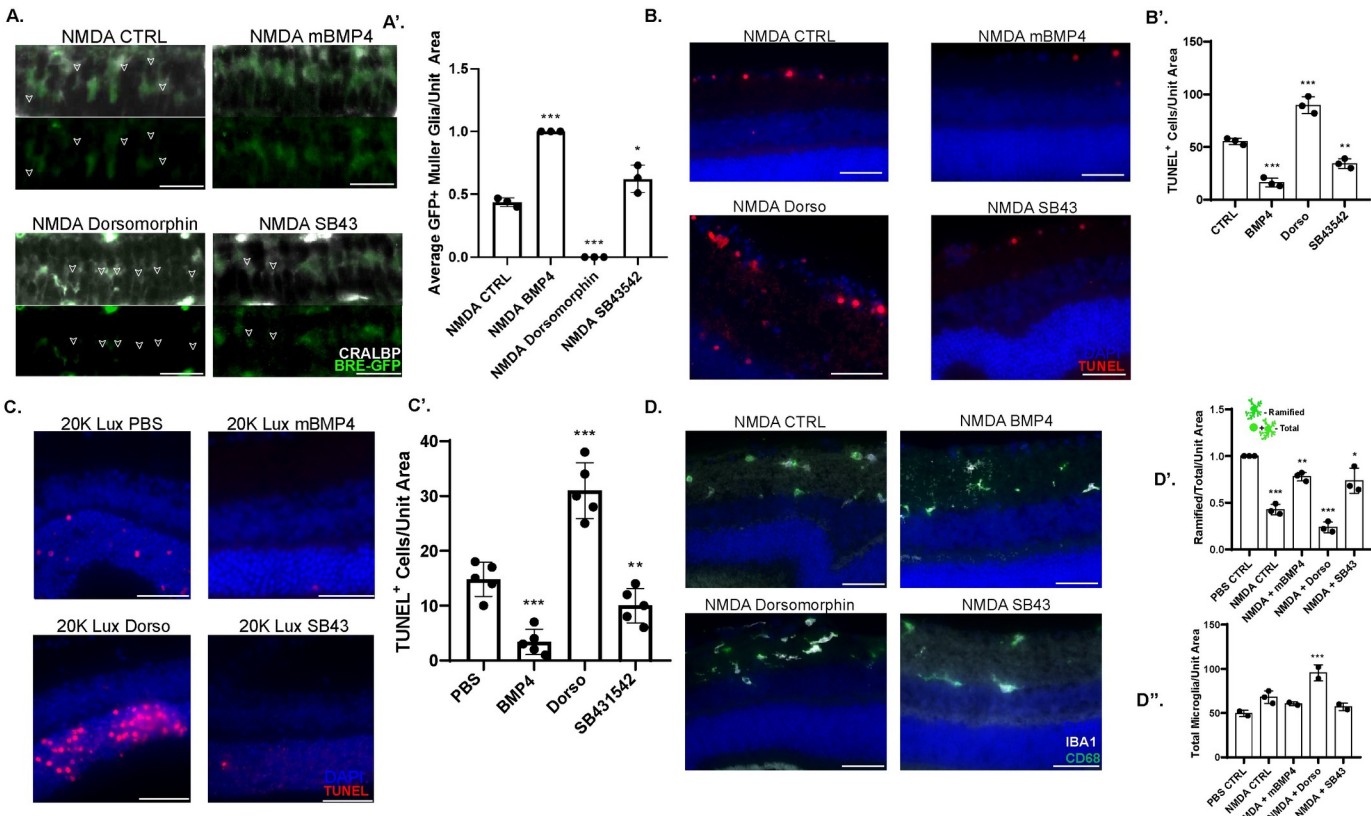

**Fig 4. Modulation of the BMP/TGFβ signaling pathway affects tissue damage induced retinal apoptosis and immune activation.** 3.5 m/o BRE-eGFP reporter mice intravitreally injected with 25mM NMDA were treated with either; PBS control, mBMP4, Dorsomorphin (BMP Inhibitor), or SB431542 (TGFβ Inhibitor). At 48 hours, BRE-GFP (green) colocalization assessed and quantified with Muller glial marker, CRALBP (white) and plotted following various treatments in comparison to control NMDA (4A-A'). TUNEL assay (red) was performed under above conditions, with total TUNEL+ cells plotted in comparison to control (4B-B'). Effects of various treatments tested in light damage conditions following 20,000 Lux exposure for 1.5 hours with retina collected 48 hours post exposure. TUNEL assay was then performed and quantified in comparison to control (4C-C'). Retinal tissue analysis for immune markers IBA1 (white) and CD68 (green), with ratio of ramified morphology immune cells over total compared between conditions (4D-D'). Total number of activated CD68+ immune cells was also compared between treatment conditions (4D-D"). Scale Bar: 20 μm (4-A). 30 μm (4B-D). Experiments were conducted with n = 2 replicates and n = 3–5 mice/condition. Error bars indicate s.e.m.; P-values from Student's t-test. *p<0.05, **p<0.01, **p<0.001. DAPI (blue) marks nuclei in all panels.

SB431542 [75] would be protective. We carried similar NMDA and light-damage experiments as above and co-treated the mice with 25 μM of SB431542 intravitreally. In both damage models, we observed that inhibition of the TGFβ pathway reduced retinal apoptosis compared to control mice (Fig 4B and 4C). Interestingly, we also observed a small reduction in BRE-GFP activity following TGFβ inhibition (Fig 4A) suggesting cross-regulation between the pathways.

## Microglia activation can be modulated by manipulating BMP/ TGFβ pathways

Lastly, we asked whether microglia activation can be biased by modulation of TGFβ and BMP pathways. Recently, morphological changes in microglia have been shown to reflect activation [76]. A change from a ramified non-activated morphology to an amoeboid shape signifies activation, and the ratio of the two morphologies can be used as a measure of relative activation. Retinal tissue was stained for the pan-microglia marker IBA1 and the inflammatory activation marker CD68. We carried out fractional analysis comparing the ratio of ramified over total (ramified and amoeboid microglia) between conditions (Fig 4D) to determine the fraction of

nonactivated microglia. Under PBS injected control conditions, all microglia have ramified morphology (S5C Fig) and NMDA damage leads to over 60% of microglia changing to amoeboid morphology (Fig 4D'). We found inhibition of BMP (Dorsomorphin treatment) led to further reduction in the ramified fraction (less than 20%), indicating increased activation, whereas supplementation of BMP4 or inhibition of TGFβ led to increases in ramified microglia, indicating reduced activation (Fig 4D'). Furthermore, the morphological changes in microglia closely correlated with CD68 expression (Fig 4D").

## Discussion

Our studies identify the morphogen Decapentaplegic (Dpp) and its mammalian homologues BMP/TGFβ as important regulators of retinal tissue survival post injury (Fig 5). Our findings suggest that in the fly, Dpp is secreted from hemocytes that previous studies have shown to be drawn to the injury site [9]. Hemocytes have also been shown to secrete Dpp in the fly intestine and embryo after injury [21,41]. We find that the response to these hemocyte derived ligands in the retina is dynamic: in Elav+ cells, Smox nuclear translocation is detected first after damage, while Mad phosphorylation occurs later and correlates with Dad::GFP expression, consistent with Mad-mediated induction of Dad, as described previously in literature [36,53,77]. Mad phosphorylation also correlates with peak levels of apoptosis, which are detected at 36 hours post UV, indicating that Mad activity is associated with the apoptotic state. Since loss of hemocyte Dpp, or of retinal Sox and Sax all result in reduced tissue loss, while loss of Tkv and Mad increased tissue loss, this suggests that hemocyte-derived Dpp induces retinal apoptosis by activating Sax/Smox signaling, while the later engagement of Tkv/Mad signaling is required to downregulate the apoptotic response. The selective engagement of Sax/Smox and Tkv/Mad signaling at different timepoints in the injury response is reminiscent of a similar dynamic observed in intestinal stem cells after bacterial infection [21,22].

As Dad is an inhibitory Smad, we hypothesize that Dad induction by Mad is required as a negative feedback signal to repress Smox signaling (S1-E). Dpp has been similarly shown to drive a proapoptotic response under damage in other tissues such as ovarian somatic cells, leg

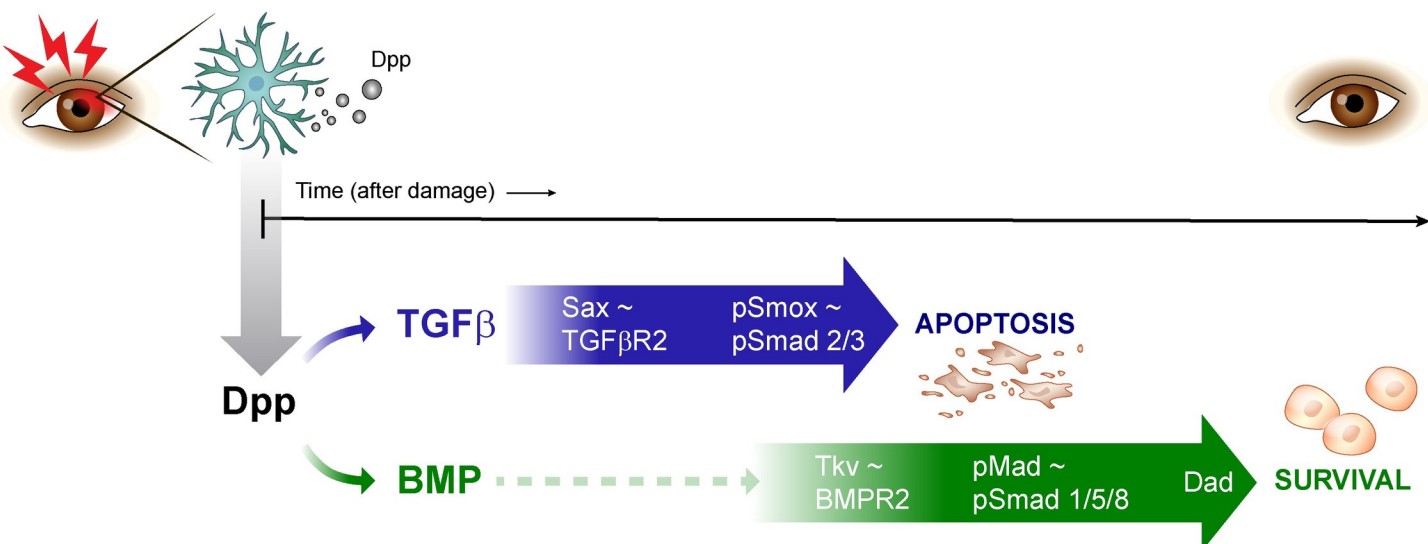

**Fig 5. Cartoon depicting the role of Dpp and TGFβ/BMP signaling as central controllers of photoreceptor death and survival after acute damage within the retina.**
Based on our data, under immune cell secreted Dpp/TGFβ/BMP ligand activation, Sax/pSmox ~ TGFβR2/pSmad2/3 activate first driving retinal apoptosis, followed by Tkv/pMad ~ BMPR2/pSmad 1/5/9 promoting increased retinal survival post-injury.

disk and retinal glia [78–80]. In ovarian cells, overexpression of Dpp was shown to drive activation of the pro-apoptotic genes *reaper* (rpr) and head involution defective (hid) [78]. Additionally, pSmox has been shown to induce cell and stage specific apoptosis of larval neurons in other *Drosophila* models [81]. Interestingly, knocking down Sax and Smox in the eye increased Dad-nGFP expression, while, as expected, knocking down Tkv and Mad significantly reduced Dad-nGFP activity. Sax/Smox signaling may thus partially repress Tkv/Mad signaling in the early phase of the injury response.

We also identified dynamic changes in the phosphorylation of the downstream effectors of the mammalian homologues to Dpp, BMP and TGFβ, in our mouse retinal stress and injury models. Under conditions of light-induced retinal stress based on our prior studies [7], we observed an initial TGFβ activation represented by early peak of pSmad2/3 followed by a peak in BMP (pSmad1/5/9) activity 6 hours later. Intriguingly, we observed an early Smox followed by a delayed Mad phosphorylation in UV-damaged flies as well, suggestive of conserved response in the arms of the pathway between the two species. These findings provide further support to homology between pSmad2/3—Smox, and Smad1/5/9—Mad, as described in other injury models in *Drosophila* as well as other species such as *Haemonchus contortus* [82–84]. We additionally observed that in our light damaged BALB/c mouse model which bears a RPE65 SNP associated with increased susceptibility to light stress [54,57], pSmad 2/3 activity exists under control conditions suggesting this arm of pathway is driven by retinal stress. Interestingly, there was no detection of Smad1/5/9 activity at any timepoint following light exposure. These results suggest that the BMP arm is protective and lack of activity of this arm of the pathway is associated with increased apoptosis.

Our data also demonstrate that knockdown of CD11b-positive immune cells in the retina significantly reduce the pSmad 1/5/9 activity in Müller glia during the light induced damage response (S5-F). This suggests that immune cells play a role in activation of the mammalian homologue of Tkv/Mad pathway, BMP. This could be by either directly secreting BMPs [85] or through intermediates by promoting BMP induced Müller glial activation [86].

Supplementation of the BMP4 and inhibition of TGFβ through the ALK inhibitor SB431542, reduce apoptosis after light and NMDA induced injury. Conversely, suppression of BMP activity through the inhibitor Dorsomorphin increases both apoptosis and total CD68 + cells, a known marker of immune cell mediated inflammation after injury [87]. Previous work has found similar findings in damaged retinal ganglion cells [27,28]. Additionally, NMDA damaged Müller glia from zebrafish, rodent and chick Müller glia derived progenitor cells (MGPCs), showing pSmad 1/5/9 promotes regeneration and pSmad 2/3 inhibits it [88–90]. Thus, our data further supports the tissue conserved response with an opportunity to modulate it to promote repair in clinical settings.

The opposing effects of apoptotic driving TGFβ/Smad2/3 and apoptosis inhibiting BMP/Smad1/5/9 on retinal injury are also correlated with different downstream target genes. BMP2/4 activity is closely associated with anti-apoptotic genes *id1/2/3*, *sox9*, *ihx2*, and *wnt 10a/11/14/7b* [91]. TGFβ is associated with TGFβ-inducible early response gene 1 (*tieg1*), Bcl-2 interacting mediator of cell death (*bim*), death-associated protein kinase (*DAP-kinase*), TGFβ induced gene human clone 3 (*bigh3*) and repression of *id2* [92–94]. Additionally, it been shown that TGFβ driven activation in injury can also drive delayed secondary necrosis in tissue, driving the cellular membrane permeable to macromolecules, inducing delayed inflammation and leading to the activation of cleaved caspase-1 (Mitchell et al., 2013) [95,96]. However, not all previous literature on TGFβ defines it as pro-apoptotic within the retina. Recent work ablating TGFβ signaling in retinal microglia using tamoxifen induced *Tgfbr2*^flox/flox mice lead to increased retinal degeneration and Muller cell gliosis [97]. Also, AAV8-TGFβ1 was recently shown to promote cone survival in rd1 mice [98]. This protective response required *Tgfbr*1 and *Tgfbr*2 activity or microglia which express

them. This suggests that while TGFβ activity downstream of *Tgfbr2 on* microglia drives secretion of pro-survival effectors, other cell types including Muller glia may have different independent roles especially in acute damage.

These results highlight the critical role of Dpp/BMP/TGFβ in regulating retinal tissue repair. Future studies targeting downstream aspects of the mammalian pathway may have therapeutic implications to delay or repair the onset of inflammation and allay retinal damage.

## Materials and methods

### *Drosophila* stocks and culture

Fly stocks were raised on standard cornmeal- and molasses-based food. All experiments were performed at 25˚C. Both sexes gave the same results in all experiments, unless otherwise described. Lines used were w;Sep::Gal4,UAS::HepACT, GMR::Gal4, GMR::Gal4;Dad-nGFP, Sep::Gal4,CD8-GFP, and GMR::Gal4;Smox::FLAG::GFP. All lines were crossed with UAS:: Dad, UAS::Dad$^{RNAi}$, UAS::Sax$^{RNAi}$, UAS::Smox$^{RNAi}$, UAS::Tkv$^{RNAi}$, UAS::Mad$^{RNAi}$, UAS:: Hml::Dpp$^{RNAi}$, or UAS::mCherry$^{RNAi}$, UAS-GFP and $W_{1118}$ as controls.

### Mice

All mice used in the described studies were housed and bred at the Association for Assessment and Accreditation of Laboratory Animal Care International accredited vivarium of the Buck Institute for Research on Aging, in a specific-pathogen-free facility, or in the UCSF Laboratory Animal Resource Center, in individually ventilated cages on a standard 12:12 light cycle. Anesthetic state was induced in mice by 1.5% isoflurane for the duration of injection experiments. In preparation for tissue collection, mice were placed in a new bedding lined cage and euthanized by displacement of air with 100% carbon dioxide for 5 minutes, and concluded with cervical dislocation to induce rapid loss of consciousness and death with a minimum of pain, distress or discomfort. All procedures were approved by the Buck Institute Institutional Animal Care and Use Committee or UCSF IACUC Animal Care Committee.

### Intraocular injections in mice

For intravitreal injection, recombinant proteins or compounds in 1-μl volume were injected into the right eye using a graduated pulled glass pipette and a wire plunger (Wiretrol II, 5-0000-2005, Drummond Scientific Company) or directly by using a Hamilton 10uL pipette following isoflurane anesthesia.

### Light and NMDA damage in mice

C57BL/6, BALB/c, or BRE-eGFP 4x backcrossed with C57BL/6 background were dark adapted for 18 hours, then intravitreally injected of control PBS or 25 mM NMDA, and recombinant protein of either; mBMP4, Dorsomorphin (BMP Inhibitor), or SB431542 (TGFβ Inhibitor). Animals were then exposed to either 5000–20,000 lux for 1–1.5 hours, or 25 mM NMDA. In case of light damage, mice were allowed to recover from anesthesia, returned to their cages, and housed in darkness until analysis. Retinal tissue was collected under control state, 0, 6, 12, 24 or 48 hours post exposure.

### UV damage in *Drosophila* pupae retina and larvae

Pupae retinas were exposed to 17.5 microJoules of UV-C radiation 24 hours post puparium formation and either collected 5 days post adulthood or 24, 27, 30, 36, and 48 hours post exposure.

## Histological analysis, imaging, and quantification methods

Retinal sections and macrophages were analyzed by immunohistochemistry (IHC) and other histological methods and imaged using a LSM 700 confocal laser-scanning microscope, images were processed sequentially on separate channels. All images were used for quantification purposes and processed with adobe photoshop software, ImageJ, Imaris 9.5.1, ZEN 3.2 and LAS X software. Eyes were fixed with 4% formaldehyde, PBS washed and placed in progressively increasing concentrations of PBS/Sucrose solution (5, 10, 15, 20%), mounted in O.C.T.(Tissue-Tek) compound in -80 degrees C overnight, sectioned into 10 micron tissue on slide, prepared as (Lamba et al., 2010) and analyzed with the following antibodies: Elav-9F8A9 (1:200, rat, DSHB), pMad-EP823Y (1:300, rabbit, abcam), DCP1-Asp216 (1:100, rabbit, Cell Signaling), pSmad 1/5/9 (1:300, rabbit, Cell Signaling), pSmad 2/3 (1:300, rabbit, Cell Signaling), GFP (1:1000, rabbit, GeneTex), CRALBP (1:200, mouse, Santa Cruz), TUNEL, IBA1 (1:200, rabbit, Abcam), and CD68 (1:100, rat, BioLegend).

## Statistical analysis

All counts are presented as average and standard error of mean (SEM). Statistical analysis was carried out using Microsoft Excel or GraphPad Prism 8.0.1, and Student's $t$ test or two-way analysis of variance (ANOVA) was used to determine statistical significance, assuming normal distribution and equal variance.

## Supporting information

**S1 Fig.** Representative images and quantitation of control pMad+ Elav+ cells under all time-points and non-UV WT genotype is compared (S1-A). Control nSmox+ Elav+ cells under all timepoints and non-UV WT genotype is compared (S1-B). GMR::Gal4;Hml::DppRNAi progeny total DCP1+ Elav+ cells is compared to control no UV animals (S1-C). Sep::Gal4; mCherry-RNAi control progeny is compared vs no UV and with GMR::Gal4 control progeny (S1-D). GMR::Gal4;UAS::Dad-RNAi is compared with control UV progeny (S1-E). Sep::Gal4; UAS-HepACT; UAS-GFP representative control image (S1-F). Scale Bar: 20 μm. Error bars indicate s.e.m.; P-values from Student's t-test. $^*$p<0.05, $^{**}$p<0.01, $^{**}$p<0.001.
(TIF)

**S2 Fig. Quantitation of Individual *Drosophila* eyes.** Representative images and quantitation of all experiments were collected below. Control UV WT progeny pMad+ Elav+ cells compared with non UV progeny (S2-A). UV exposed control nSmox+ Elav+ cells is compared with non UV progeny (S2-B). Total WT DCP1+ Elav+ cells quantitation compared with no UV (S2-C). Total Dad-nGFP+ Elav+ cells compared between UV and no UV WT progeny (S2-C'). Individual quantitation of total GMR::G4;Hml::Dpp$^{RNAi}$ DCP1+ Elav+ cells and Dad-nGFP+ Elav+ cells compared with WT UV (S2-D). Error bars indicate s.e.m.; P-values from Student's t-test. $^*$p<0.05, $^{**}$p<0.01, $^{**}$p<0.001.
(TIF)

**S3 Fig. Quantitation of Individual *Drosophila* eyes continued.** Representative images and quantitation of all experiments were collected below. UV exposed GMR::G4;SaxRNAi individual progeny DCP1+ Elav+ cells and Dad-nGFP+ Elav+ cells are compared to UV WT (S3-A). UV exposed GMR::G4;SmoxRNAi individual progeny DCP1+ Elav+ cells and Dad-nGFP + Elav+ cells are compared to UV WT (S3-B). UV exposed GMR::G4;TkvRNAi individual progeny DCP1+ Elav+ and Dad-nGFP+ Elav+ cells are compared to UV WT (S3-C). UV exposed GMR::G4;MadRNAi individual progeny DCP1+ Elav+ and Dad-nGFP+ Elav+ cells are compared to UV WT (S3-D). Scale Bar: 20 μm. Error bars indicate s.e.m.; P-values from

Student's t-test. $^*p<0.05$, $^{**}p<0.01$, $^{**}p<0.001$.
(TIF)

**S4 Fig. UV exposed Sep::G4;Smox and MadRNAi individual progeny DCP1+ Elav+ cells to UV Sep WT.** Error bars indicate s.e.m.; P-values from Student's t-test. $^*p<0.05$, $^{**}p<0.01$, $^{**}p<0.001$.
(TIF)

**S5 Fig.** pSmad 1/5/9 (BMP) and pSmad 2/3 (TGFB) control images of C57 animals (S5-A). TUNEL stain in red of BALB/c animals post light exposure, with DAPI in blue (S5-B). PBS WT Control unexposed retina with IBA1 in white and CD68 in green (S5-C). pSmad 1/5/9 representative images of BALB/c animals post light exposure at 0, 24 and 36 hours (S5-E). Representative images of DT injected CD11b::DTR mice with pSmad 1/5/9 (BMP) in red, and CRALBP in white (S5-E). Scale Bar: 30 μm.
(TIF)

## Acknowledgments

The authors thank members of the Lamba and Jasper labs for advice and technical assistance. We would like to thank the UCSF Vision Core for their support.

## Author Contributions

**Conceptualization:** Joshua Kramer, Heinrich Jasper, Deepak A. Lamba.

**Formal analysis:** Joshua Kramer, Heinrich Jasper, Deepak A. Lamba.

**Funding acquisition:** Heinrich Jasper, Deepak A. Lamba.

**Investigation:** Joshua Kramer, Joana Neves, Mia Koniikusic.

**Methodology:** Joana Neves.

**Resources:** Heinrich Jasper, Deepak A. Lamba.

**Supervision:** Heinrich Jasper, Deepak A. Lamba.

**Validation:** Joshua Kramer.

**Writing – original draft:** Joshua Kramer.

**Writing – review & editing:** Joshua Kramer, Heinrich Jasper, Deepak A. Lamba.

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
