## [Decision Letter · Decision Letter 0]

15 Jul 2021

PONE-D-21-17742

Dpp/TGFβ-superfamily play a dual conserved role in mediating the damage response in the retina

PLOS ONE

Dear Dr. Lamba,

Thank you for submitting your manuscript to PLOS ONE. After careful consideration, we feel that it has merit but does not fully meet PLOS ONE’s publication criteria as it currently stands. Therefore, we invite you to submit a revised version of the manuscript that addresses the points raised during the review process.

Most reviewer comments refer to data clarity or interpretation. PLOSOne publishes clear-cut data (including well described experiments and appropriate controls), without emphasizing impact. Please address the concerns of reviewers, by reporting the missing information, adjusting your claims/statements to be in sync with the data, and/or rephrasing appropriately. In particular, please comment on the choice of controls in response to reviewer 1.

We look forward to receiving your revised manuscript.

Kind regards,

Tudor C Badea, M.D., M.A., Ph.D.

Academic Editor

PLOS ONE

Journal Requirements:

3. In your Methods section, please provide additional information on the animal research and ensure you have included details on : (a) methods of sacrifice (b) methods of anesthesia and/or analgesia, and (c) efforts to alleviate suffering

4. Thank you for stating the following in the Financial Disclosure section:

"The research presented here is supported by NIH grants (R01 EY025779 and

EY032197 to DL; AG057353 and EY018177 to HJ; P30 Vision Core grant to UCSF

Dept of Ophthalmology), and the Research to Prevent Blindness (unrestricted grant to

UCSF Dept of Ophthalmology). The funders had no role in study design, data

collection and analysis, decision to publish, or preparation of the manuscript."

We note that one or more of the authors are employed by a commercial company: "Genentech, Inc."

5. Please include a caption for figure 5.

Additional Editor Comments (if provided):

Most reviewer comments refer to data clarity or interpretation. PLOSOne publishes clear-cut data (including well described experiments and appropriate controls), without emphasizing impact. Please address the concerns of reviewers, by reporting the missing information, adjusting your claims/statements to be in sync with the data, and/or rephrasing appropriately.

Reviewers' comments:

Reviewer's Responses to Questions

**Comments to the Author**

1. Is the manuscript technically sound, and do the data support the conclusions?

Reviewer #1: Yes

Reviewer #2: Yes

2. Has the statistical analysis been performed appropriately and rigorously? 

Reviewer #1: Yes

Reviewer #2: Yes

3. Have the authors made all data underlying the findings in their manuscript fully available?

Reviewer #1: Yes

Reviewer #2: Yes

4. Is the manuscript presented in an intelligible fashion and written in standard English?

Reviewer #1: Yes

Reviewer #2: Yes

5. Review Comments to the Author

Reviewer #1: This is paper “Dpp/TGFβ-superfamily play a dual conserved role in mediating the damage response in the retina” is a nice work by Kramer and colleagues. I have a few concerns to be addressed before it is final for publication.

Major comments

1. Autors say “Tkv turnover is reduced during ISC activation, allowing for Tkv protein accumulation and replacement of Sax in Tkv/Punt complexes(22).”

Reduced Tkv turnover would prevent its accumulation.

2. Autors say “Overexpression of the negative feedback inhibitor of Dpp signaling, Daughters against Dpp (Dad; (36,37)), also….”

Similar to knockdown experiment, ideal control for overexpression(UAS-GFP/mcherry) should also have been used.

3. Autors say “These kinetics are consistent with the hypothesis that Mad activation induces Dad transcription to inhibit Sax/Smox signaling in the eye, thus limiting apoptosis.”

Not necessarily implied by the result. In Figure D’ and E’’’, nsmox level continues to increase, even when dad levels decrease(from 27h to 30h)

4. In Figure 2, all the labels specify usage of WT, whereas it was mentioned earlier that control for knockdown experiment had UAS- mCherry RNAi. Labels have to be changed accordingly.

5. Autors say “These studies suggest that TGFβ signaling may be involved in the early inflammatory response, while BMPs may play a role in a later protective anti-apoptotic pro-repair response which BALB/c mice lack.”

Here, mere absence of BMP in Balb/c doesn’t imply its anti apoptotic/pro-repair pathway. It needn’t be the cause, it could be an effect.

Minor comments

1. There are typo errors in the manuscript, like in TGFβ, the symbol is replaced by a square at some places. May be a computer error too.

2. Some of the data need further clarification, like in the statement ‘These studies suggest that TGFβ signaling may be involved in the early inflammatory response, while BMPs may play a role in a later protective anti-apoptotic pro-repair response which BALB/c mice lack.’, the results are based on smad2/3 or smad1/5/9 nuclear localization only. If they have to prove the real role of these pathways in the stated contexts then showing with pharmacological inhibitors would be a better choice.

3. Image quality need to be improved, everything including text is so blurred that I cannot conclude anything after looking at them.

4. Protective effect of TGFβ inhibition on apoptosis is shown by sb431542 treatment, it would be better to show its flip side by overexpression of TGFβ signaling as shown in the BMP pathway.

Reviewer #2: In this manuscript the authors elegantly show using a Drosophila model for UV-induced retinal damage, that Dpp released from immune cells promotes tissue loss after UV-induced retinal damage. Interestingly they found that, Dpp-mediated stimulation of Saxophone / Smox signaling promotes apoptosis, while at a later stage, stimulation of the Thickveins / Mad axis promotes tissue repair and survival.

This seems to be a key evolutionary conserved mechanisms by which retinal homeostasis is maintained.

Critique

1. Figure 5 does not have a Figure legend.

2. Is unclear from the Figure Legends how many times the experiments were repeated and if they were run in duplicates. This should be introduced in the Figure Legend.

3. In Figure 1 A+ and B+ will benefit from larger font. This is valid for Figure 2.

6. PLOS authors have the option to publish the peer review history of their article (what does this mean?). If published, this will include your full peer review and any attached files.

Reviewer #1: No

Reviewer #2: No

---

## [Author Response · Author response to Decision Letter 0]

17 Sep 2021

Dear Editors,

We thank the reviewers for their comments on the manuscript and have edited the manuscript to address their concerns. 

Editorial Comments:

>We have checked and ensured the submission meets the naming and style guidelines.

 2. Please review your reference list to ensure that it is complete and correct. 

>One paper was revised and that reference is used instead of the original. Otherwise all references are acceptable under PLOS ONE guidelines.

 3. In your Methods section, please provide additional information on the animal research and ensure you have included details on : (a) methods of sacrifice (b) methods of anesthesia and/or analgesia, and (c) efforts to alleviate suffering

>We have updated the Methods to include the requested details on animal handling and euthanasia. 

4. Question regarding Financial Disclosure and Competing Interests for Dr. Jasper w.r.t. Genentech affiliation.

>Updated funding and competing interest statements are below.

Funding Statement: The research presented here is supported by NIH grants (R01 EY025779 and EY032197 to DL; AG057353 and EY018177 to HJ; P30 Vision Core grant to UCSF Dept of Ophthalmology), and the Research to Prevent Blindness (unrestricted grant to UCSF Dept of Ophthalmology). The funders had no role in study design, data collection and analysis, decision to publish, or preparation of the manuscript. Dr. Jasper’s affiliation with Genetech, Inc provided support in the form of salaries for author HJ, but did not have any additional role in the study design, data collection and analysis, decision to publish, or preparation of the manuscript. The specific roles of these authors are articulated in the ‘author contributions’ section.

Competing Interests Statement: I have read the journal's policy and the following authors of this manuscript have the following competing interests: Heinrich Jasper is an employee of Genentech Inc. This commercial affiliation does not alter author’s adherence to all PLOS ONE policies on sharing data and materials.

5. Please include a caption for figure 5.

>We apologize for the oversight and have now included a legend for the figure.

Review Comments to the Author

Reviewer #1: This is paper “Dpp/TGFβ-superfamily play a dual conserved role in mediating the damage response in the retina” is a nice work by Kramer and colleagues. I have a few concerns to be addressed before it is final for publication.

Major comments

1. Autors say “Tkv turnover is reduced during ISC activation, allowing for Tkv protein accumulation and replacement of Sax in Tkv/Punt complexes(22).” Reduced Tkv turnover would prevent its accumulation.

>We apologize for any confusion this statement caused. Our previous study cited here (#22, Cai et al 2019) showed that after ISC activation, Tkv degradation is reduced, resulting in accumulation of Tkv protein. We used Tkv turnover as synonymous with degradation, but understand that this can be misinterpreted. We now use the term ‘degradation’ instead. 

2. Autors say “Overexpression of the negative feedback inhibitor of Dpp signaling, Daughters against Dpp (Dad; (36,37)), also….” Similar to knockdown experiment, ideal control for overexpression(UAS-GFP/mcherry) should also have been used.

>As requested, we have carried out that study and included the UAS-GFP control experimental data in Figure 1A, with representative image in S1F. We do not observe any significant differences between the UAS-GFP and UAS::mCherry-RNAi lines.

3. Autors say “These kinetics are consistent with the hypothesis that Mad activation induces Dad transcription to inhibit Sax/Smox signaling in the eye, thus limiting apoptosis.” 

Not necessarily implied by the result. In Figure D’ and E’’’, nsmox level continues to increase, even when dad levels decrease(from 27h to 30h)

>We believe that the kinetics of Dad induction and Smox nuclear translocation are linked, but the readouts are not entirely comparable. With other words, the induction of Dad was measured using a GFP reporter line, which is likely to lag true Dad protein production (GFP fluorescence requires about 2 hours to be visible). We measure Smox nuclear translocation in real time, however. We therefore interpret the results as showing that while Dad is being induced between 27 and 30 (in the UV treated samples), nuclear Smox levels are decreasing. In the untreated samples, the relationship between Dad and nuclear smox is less clear, but the expression of Dad is also much lower than in the UV-exposed samples. 

To clarify our thinking, we have now edited the sentence to the following: “These kinetics led us to hypothesize that Mad activation induces Dad transcription, resulting in inhibition of Smox nuclear translocation after UV exposure, limiting apoptosis.”

4. In Figure 2, all the labels specify usage of WT, whereas it was mentioned earlier that control for knockdown experiment had UAS- mCherry RNAi. Labels have to be changed accordingly.

>We have changed labels to illustrate that UAS-mCherry RNAi was used for control.

5. Autors say “These studies suggest that TGFβ signaling may be involved in the early inflammatory response, while BMPs may play a role in a later protective anti-apoptotic pro-repair response which BALB/c mice lack.”

Here, mere absence of BMP in Balb/c doesn’t imply its anti apoptotic/pro-repair pathway. It needn’t be the cause, it could be an effect.

> The reviewer here raises a good point. We have now edited the statement to the following ”These studies suggest that TGFβ signaling may be involved in the early response which is typically of inflammatory nature, while BMPs may play a role in a later response which tends to be protective and anti-apoptotic. The absence of BMP/SMAD1/5/9 response in BALB/c mice was further investigated below to test a cause or effect conundrum.” 

Additionally, experiments further in the manuscript answer this question using recombinant protein and pharmacological inhibitor injections in mice.

Minor comments

1. There are typo errors in the manuscript, like in TGFβ, the symbol is replaced by a square at some places. May be a computer error too.

>We did not find β symbol to square errors in our uploaded version of the manuscript and was likely an issue with PDF conversion. All miscellaneous typos were corrected.

2. Some of the data need further clarification, like in the statement ‘These studies suggest that TGFβ signaling may be involved in the early inflammatory response, while BMPs may play a role in a later protective anti-apoptotic pro-repair response which BALB/c mice lack.’, the results are based on smad2/3 or smad1/5/9 nuclear localization only. If they have to prove the real role of these pathways in the stated contexts then showing with pharmacological inhibitors would be a better choice.

> Since Smad 2/3 and Smad 1/5/9 are canonical downstream effectors for TGF and BMP signaling respectively, it is reasonable to hypothesize their role at early and late state of tissue response based on their activation and nuclear localization. To further clarify this, we have edited the statement to the following ”These studies suggest that TGFβ signaling may be involved in the early response which is typically of inflammatory nature, while BMPs may play a role in a later response which tends to be protective and anti-apoptotic. The absence of BMP/SMAD1/5/9 response in BALB/c mice was further investigated below to test a cause or effect conundrum.”

The pharmacological manipulations are described further in the manuscript.

3. Image quality need to be improved, everything including text is so blurred that I cannot conclude anything after looking at them.

> All uploaded images passed check. The Image quality in the preview PDF is poor quality. However, full figure images on the links in the PDF are of high resolution.

4. Protective effect of TGFβ inhibition on apoptosis is shown by sb431542 treatment, it would be better to show its flip side by overexpression of TGFβ signaling as shown in the BMP pathway.

> That is an excellent suggestion and obvious next step. However, due to strong effects of TGFβ on vasculature, those studies require careful consideration and will be carried out in future studies.

Reviewer #2: In this manuscript the authors elegantly show using a Drosophila model for UV-induced retinal damage, that Dpp released from immune cells promotes tissue loss after UV-induced retinal damage. Interestingly they found that, Dpp-mediated stimulation of Saxophone / Smox signaling promotes apoptosis, while at a later stage, stimulation of the Thickveins / Mad axis promotes tissue repair and survival.

This seems to be a key evolutionary conserved mechanisms by which retinal homeostasis is maintained.

Critique

1. Figure 5 does not have a Figure legend.

>We apologize for the oversight and have now included a legend for the figure.

2. Is unclear from the Figure Legends how many times the experiments were repeated and if they were run in duplicates. This should be introduced in the Figure Legend.

> The requested details have now been included.

3. In Figure 1 A+ and B+ will benefit from larger font. This is valid for Figure 2.

>We have increased the font size as requested.

---

## [Editor Report · Decision Letter 1]

7 Oct 2021

Dpp/TGFβ-superfamily play a dual conserved role in mediating the damage response in the retina

PONE-D-21-17742R1

Dear Dr. Lamba,

We’re pleased to inform you that your manuscript has been judged scientifically suitable for publication and will be formally accepted for publication once it meets all outstanding technical requirements.

Kind regards,

Tudor C Badea, M.D., M.A., Ph.D.

Academic Editor

PLOS ONE
---

## [Editor Report · Acceptance letter]

18 Oct 2021

PONE-D-21-17742R1 

Dpp/TGFβ-superfamily play a dual conserved role in mediating the damage response in the retina 

Dear Dr. Lamba:

I'm pleased to inform you that your manuscript has been deemed suitable for publication in PLOS ONE. Congratulations! Your manuscript is now with our production department. 

Kind regards, 

on behalf of

Dr. Tudor C Badea 

Academic Editor

PLOS ONE